# The Evolutionary Game Analysis of Low Carbon Production Behaviour of Farmers, Government and Consumers in Food Safety Source Governance

**DOI:** 10.3390/ijerph191912211

**Published:** 2022-09-26

**Authors:** Yayan Xie, Yang Su, Feng Li

**Affiliations:** 1College of Economics and Trade, Xinjiang Agricultural University, Urumqi 830052, China; 2School of Business Administration, Xinjiang University of Finance and Economics, Urumqi 830012, China

**Keywords:** incentive compatibility theory, food security source governance, evolutionary game, farmers, low carbon production technology

## Abstract

Whether the quality of agricultural products is safe or not is related to issues of food safety and low carbon production in agriculture. Based on evolutionary game theory, this paper establishes a game model among government, farmers and consumers and analyzes the dynamic evolutionary process and evolutionary stable strategies of the major stakeholders. The results show that: (i) government subsidy coefficient, farmers’ penalty coefficient for not producing, consumer trust coefficient and willingness to pay carbon labelled agricultural products premium are positively related to farmers’ adoption of low-carbon production behaviour, and fraud penalty coefficient and farmers’ cost of adopting low-carbon production technology are negatively related; (ii) farmers’ sensitivity to government regulation policies is: fraud penalty coefficient = farmers’ cost of adopting low-carbon production technology > government subsidy The sensitivity of farmers to government regulation policies is: fraud penalty coefficient = cost of low-carbon production technology > government subsidy > penalty coefficient for non-production, and the sensitivity of farmers to direct market stimulation is: consumer trust coefficient > coefficient of willingness to pay premium for carbon labelled agricultural products, and the additional benefit coefficient has no significant effect on farmers’ decision-making; (iii) In the early stage, the source control of food safety mainly depends on the government’s policy intervention. In the later stage, the establishment of carbon label agricultural products market incentive mechanism can achieve long-term stable and effective source control of food safety.

## 1. Introduction

Food safety is closely related to public health and is an important part of public health safety [1]. Among them, foodborne diseases refer to infectious and toxic diseases caused by pathogenic factors in food entering the human body. Foodborne diseases are not only one of the most widespread public health problems in the world, but also the primary focus of food safety issues [2]. There are one billion cases of foodborne diseases in the world every year, about 48 million in the United States [3], about 11 million in Canada [4], and 200 million in China [5]. In order to cope with the serious burden of foodborne diseases on human public health, all countries have strengthened the source control of food safety.

Agricultural production is the starting point and source of the food lifecycle, and source control is the key to ensuring food safety [6]. At the same time, the high intensity of agricultural chemical application in China makes food safety governance at the source relatively tricky [7]. General Secretary Xi Jinping has repeatedly emphasized that “the source of food safety is in agricultural products, the foundation is in agriculture, and we must correct the root cause and first grasp the quality of agricultural products” In grasping the quality and safety of agricultural products, it is also necessary to effectively accelerate the pace of low-carbon transformation of agricultural production [8]. Therefore, the People’s Republic of China’s national economic and social development of the fourteenth five-year plan and 2035 vision outline further pointed out to “promote the construction of food safety assurance project to attack the action” and stressed that “the implementation of pesticide and veterinary drug use reduction and the origin of environmental clean-up action“. In addition [9], on 17 September 2021, the National Health Commission issued 17 new national food safety standards and one revised list, including 1000 standards for pesticide residue limits [10]. This shows that the low-carbon production behavior of farmers has become an important part of food safety governance.

Farmers are the direct implementers of agricultural production, and the safety of their production determines the possibility of food safety, which directly affects the general situation of food quality and safety control and stable social development [11]. Agricultural low-carbon production refers to a new type of agricultural production mode that makes full use of technology, policies and management measures to achieve the sustained growth of agricultural output while reducing agricultural material input and agricultural pollution as much as possible [12]. Chemical fertilizers and pesticides are important pollution sources of agricultural production. Reducing the use of chemical fertilizers and pesticides is an important manifestation of the low-carbon agricultural production behavior of farmers, which is objectively conducive to the ultimate realization of the development goal of food safety management. However, compared to traditional agricultural production methods, low-carbon production techniques require higher labor inputs and more stringent operational requirements [13]. As “rational economic people”, farmers will inevitably consider costs, benefits, and risks, which will lead to differences in the degree of application of low-carbon production technologies among farmers [14]. How do we promote the adoption of low-carbon production technologies among farmers? How can we coordinate the efforts of all parties to actively participate in the construction of food safety governance at the source? Rational answers to the synergistic interests of the three actors in food safety source governance will facilitate the development of more effective food safety source governance policies and enhance the ecological and economic benefits of agriculture.

Scholars have researched food safety governance sources, mainly from single-actor stakeholder motivation and multi-actor evolutionary games. The single stakeholder motivation focuses on maximizing the interests of farmers, government, and consumers and the factors that influence them. First, from the farmer’s perspective, the agricultural market is similar to the lemon market, and farmers will act opportunistically to maximize profits and thus create moral hazard problems [15]. Measures such as vertical integration [16], increasing the production costs of adverse selection and education [17] and training for farmers can promote the adoption of low-carbon production technologies by farmers [18]. The second is the government’s pursuit of social welfare maximization in food safety at source, the use of the “visible hand” to regulate the market [19], and the establishment of a legal system for agricultural quality and safety (punishment and integrity mechanisms) [20], the integration and coordination of government departments in supervision [21] and the construction of a guarantee system (standardization, certification system, traceability system, early warning system) [22] are considered as effective measures for food safety governance at source; third, consumers pursue personal utility maximization in the agricultural market and attributes such as greenness and safety of carbon labeled agricultural products are organic components of consumer utility [23], and awareness of agricultural quality and safety [24], urban-rural disparity [25], education level [26], food safety information disclosure [27], and income disparity [28], are considered to be influential factors affecting consumers’ purchase of carbon labeled agricultural products.

Secondly, the study of food safety governance source from the perspective of a multi-subject evolutionary game, mainly from the perspective of “government-farmers (agribusiness)” or “government-consumers”, explores the game relationship of food safety governance. In the “government-farmers (agribusiness)” game, Lixia Liu et al. analyzed the evolutionary game from the perspectives of agribusiness, government, and farmers. They concluded that reasonable government subsidies and carbon taxes for agribusiness and farmers could increase the motivation of agribusiness and farmers to participate in low-carbon agriculture, thus ensuring food safety governance at the source [29]. Ben Henderson et al. argue that A global carbon tax on AFOLU is found to be twice as effective in lowering emissions as an equivalently priced emission abatement subsidy because the latter keeps high emitting producers in business [30]. Thus, it can be seen that there is no consensus on the effect of government policies on farmers (agribusinesses), and the effect of policy instruments is controversial. In terms of the “government-consumer” game, Y TENG et al. concluded that improving the ability to identify high-quality agricultural products and choosing formal sales channels to purchase high-quality agricultural products directly affects the construction of food traceability systems and consumer confidence in food safety [31]. Teresa Del Giudice et al. found that the degree of consumer preference for green agricultural products can influence the construction of green agricultural markets [32]. In summary, studies on food safety governance at source from a game perspective are mostly related to the government’s regulatory system for farmers (agribusinesses), food safety standards, traceability system construction, and consumers’ carbon labeling preferences, but less on the dynamic game among government, farmers, and consumers. The game among the three stakeholders is the reason for farmers’ difference in the adoption of low carbon production technology.

However, what is the direction of incentives for farmers (agribusinesses) to adopt low-carbon production behaviors in food safety at the source by policy instruments and market incentives? Are there differences in the speed and direction of farmers’ response (agribusinesses) to different types of policy regulation and market incentives at different intensity levels? If so, how do farmers prioritize different policy instruments and market incentives in food safety management at source? How do farmers’ and consumers’ behavioral decisions respond over time to implementing policy instruments and market incentives?

In order to answer the above questions, it is necessary to conduct a sensitivity analysis on the strategy choice of each subject under policy regulation and market dispatch by establishing the dynamic replication equations of each of their actors, computing the stable evolutionary strategies of the three parties, and then assigning and simulating each variable through Matlab2021b to explore the key factors affecting the production behavior of farmers. In order to grasp the intrinsic mechanism of farmers’ production behavior at the source, which helps to promote food safety source management and low-carbon transformation of agriculture.

## 2. Theoretical Analytical Framework: Analysis of Incentive-Compatible Mechanisms for Food Safety Governance at Source

Intensive agricultural production method makes food safety governance relatively difficult from the source [33], mainly due to the conflict of interests among three stakeholders: farmers, government, and consumers. Agricultural products have the attribute of trust goods [34], and consumers and government are on the information-deficient side. To prevent hidden behaviors of farmers [35], an effective incentive-compatible mechanism for food safety governance at the source needs to be constructed. As shown in Figure 1.

### 2.1. Farmer-Government Incentive Compatibility Mechanism in Food Safety Governance at Source

In food security source management, the government can regulate farmers’ low-carbon production behaviour through subsidies and penalties [36]. On the one hand, the government promotes low-carbon production subsidies and penalises farmers who maintain traditional production behaviours, which provides some positive incentives for farmers to adopt low-carbon production, but also raises the moral risk of farmers’ adverse selection [37], and due to the hidden characteristics of the agricultural market [38], farmers may generate fraudulent subsidies based on the principle of maximising personal interests, thus avoiding punishment [39]. On the other hand, the government has increased supervision and punished farmers for fraudulent subsidies, which has curbed the spread of farmers’ fraudulent subsidies to a certain extent, but the illegal use of environmentally polluting chemicals by farmers enjoying subsidies can increase the output of agricultural products to reduce the unit cost and rapidly occupy the agricultural market [40], resulting in the market phenomenon of “bad money expelling good money”. Overall, the government’s influence on farmers’ low-carbon production behaviour will depend on the impact of policy strength on farmers’ individual expected returns.

### 2.2. Farmer-Consumer Incentive Compatibility Mechanism in Food Safety Governance at Source

In food safety governance at source, consumers’ willingness to pay a premium for carbon-labeled agricultural products can moderate farmers’ low-carbon production behavior [41]. First, consumers’ failure to effectively identify the safety of food products gives a considerable market for less safe agricultural products due to price advantages [42]. The establishment of carbon label agricultural brands and markets highlights the differentiation of safe agricultural products, reduces the hidden characteristics of information asymmetry of agricultural products, and in turn, increases consumers’ willingness to pay for carbon label agricultural premiums [43]; Secondly, due to the improvement of the living standard of the whole society, the goal of farmers’ production activities gradually changed from “high yield” to “high quality”. Some farmers started to produce carbon-labeled agricultural products [44]. Overall, the influence of consumers on farmers’ low-carbon production behavior will depend on consumers’ willingness to pay for carbon-labeled agricultural products premium on farmers’ individual expected returns.

### 2.3. Government-Consumer Incentive Compatibility Mechanism in Food Safety Governance at Source

In food safety source governance, increased government investment in food safety source governance leads to consumer trust in carbon-labeled agricultural products. This consumer trust can propel food safety source governance into a virtuous cycle [45]. On the one hand, the government guides farmers to engage in low-carbon production and establish a market for carbon-labeled agricultural products; on the other hand, consumers regain trust in the government’s governance of food safety issues and restore the attributes of agricultural products as trust goods, and consumers’ trust in the government’s environmental governance helps increase market demand for carbon-labeled agricultural products, which in turn drives farmers to adopt low-carbon production technologies and provide more high-quality primary agricultural products to obtain consumer benefits from low-carbon preferences [46]. Overall, consumer trust and government construction of carbon-labeled agricultural products market positively affect farmers’ individual expected returns.

Based on the above analysis, this paper proposes the following research hypotheses.

**Hypothesis** **1.**
*The three policy regulation tools of government low-carbon production subsidy, farmer non-production penalty, and farmer fraud penalty have a dual role of promoting and inhibiting farmers’ low-carbon production, and when the intensity of these three policy regulations reaches a specific value, the direction of the incentive effect on farmers’ low-carbon production turns from positive to negative.*


**Hypothesis** **2.**
*The greater the three market incentives of consumer trust, the additional benefit coefficient of the government’s carbon-labeled agricultural market, and the willingness to pay the premium for carbon-labeled agricultural products, the greater the probability that farmers will adopt low carbon production technologies, indicating that consumer trust, the additional benefit coefficient of the government’s carbon labeled agricultural market, and the willingness to pay the premium for carbon labeled agricultural products have a positive effect in food safety source management.*


## 3. Materials and Methods—Evolutionary Game Model

Based on the problem description in the above theoretical analysis. The materials and methods in the third part of this paper next begin with model assumptions for the evolutionary game model constructed in this paper.

### 3.1. Model Assumption

A_1_ represents the three parties of the game are the government, farmers, and consumers, and the three parties are in the initial stage of the evolutionary game model without considering other influencing factors.

A_2_ represents the government has two strategies, namely ‘subsidising’ or ‘not subsidising’. When the government chooses a ‘subsidy’ strategy, it subsidises the adoption of low-carbon production practices by farmers, denoted by A. The subsidy factor is θ, and the amount of government subsidy is θA. and the government inspects farmers for fraudulent subsidies at a cost of Q. The farmer has probability μ of committing fraudulent acts, and if the fraud is committed, then the government must inspect and penalise the farmer E. The fraud penalty factor is η, then the fraud penalty is ηE, and the farmer is ordered to adopt low-carbon production practices. When the government chooses the “subsidy” strategy or not, the farmer is penalised for not adopting the low cost of production D, with a penalty coefficient of γ, the farmer is penalised with γD. and 0 ≤ θ, η, μ, γ ≤ 1; when the government chooses the “no subsidy” strategy When the government chooses the “no subsidy” strategy, the government has no power to provide incentives to farmers.

A_3_ represents farmers have two strategies, namely ‘adoption’ or ‘non-adoption’ of low carbon production practices. When farmers adopt, they incur the cost of adopting low-carbon production $C_{1}$ bringing in revenue from the sale of carbon-labelled produce M_1_, while the government receives the low-carbon social benefits of farmers’ adoption of low-carbon production behaviour $F_{1}$; When farmers do not adopt low-carbon production behavior then ordinary sales revenue $M_{2}$, loss of carbon label safe agricultural market revenue $C_{2}$, the government at this time to obtain the general social benefits $F_{2}$, and the government at this time whether or not to choose the “subsidy” strategy, will invest in building carbon label agricultural market P, the consumer trust coefficient is β, and 0 ≤ β ≤ 1.

A_4_ represents consumers have two strategies, namely “buy” or “don’t buy”. When the consumer chooses the “buy” strategy, the benefit of carbon labelled produce for the consumer is $M_{3}$, the government’s carbon labelled produce input brings additional environmental benefit B to the consumer, and the additional benefit factor is φ. The additional environmental benefit for the consumer is Bφ, and the purchase brings additional carbon labelled produce benefits r to the farmer, and the coefficient of influence of willingness to pay carbon labelled produce premium on benefits is ρ. The amount of benefits to the farmer is rρ; When the consumer “does not buy”, the consumer receives an ordinary benefit of M_4_. and 0 ≤ φ, ρ ≤ 1. The corresponding parameters are shown in Table 1.

### 3.2. Construction of the Revenue Matrix

The government, farmers, and consumers will make strategy choices according to their wishes. The probability that the government chooses the subsidy strategy is x, the probability that it chooses the unsubsidized strategy is 1 − x, and the probability that farmers choose the adoption strategy is y. Vice versa is 1 − y, the probability that consumers choose the purchase strategy is z, and vice versa is 1 − z. And x, y, z [0, 1]. Based on this, a game model of agricultural market circulation based on an evolutionary game was established. A government, farmer, and consumer payoff matrix were constructed based on the above assumptions. As shown in Table 2 and Table 3 below.

### 3.3. Construction of the Expectation Function of the Revenue of the Three Parties

#### 3.3.1. Benefit Expectations from the Government Perspective

Let the expectation of return when the government is subsidizing be $U_{11}$, the expectation of return when the government is not subsidizing be $U_{12}$, and the average expectation of return be $\bar{U_{1}}$, as follows.
(1)$$\begin{array}{l} U_{11}=\mathrm{yz}\left(\mu \eta Ε-Q-\theta A+\left.F_{1}\right)\right.+y\left(1-\left.z\right)\right.\left(\mu \eta Ε-Q-\theta A+\left.F_{1}\right)\right.+z\left(1-\left.y\right)\left(\gamma D+F_{2}-\left.Ρ\beta \right)\right.\right.+\left(1-\left.y\right)\right.\left(1-\left.z\right)\right.\left(\gamma D+F_{2}-\left.Ρ\beta \right)\right.\\ =y\left(\mu \eta Ε-Q-\theta A+\left.F_{1}\right)\right.+\left(1-\left.y\right)\right.\left(\gamma D+F_{2}-\left.Ρ\beta \right)\right. \end{array}$$
(2)$$\begin{array}{l} U_{12}=yzF_{1}+y\left(1-z\right)F_{1}+z\left(1-y\right)\left(F_{2}-P\beta \right)+\left(1-y\right)\left(1-z\right)\left(F_{2}-P\beta \right)\\ =yF_{1}+\left(1-y\right)\left(F_{2}-P\beta \right) \end{array}$$
(3)$$\begin{array}{l} \bar{U_{1}}=U_{11}x+U_{12}\left(1-x\right)\\ =yx\left(\mu \eta Ε-Q-\theta A+F_{1}\right)+\left(1-y\right)x\left(\gamma D+F_{2}-P\beta \right)+y\left(1-x\right)F_{1}+\left(1-y\right)\left(1-x\right)\left(F_{2}-P\beta \right)\\ =yx\left(\mu \eta Ε-Q-\theta A\right)+\left(1-y\right)x\gamma D+yF_{1}+\left(1-y\right)\left(F_{2}-P\beta \right) \end{array}$$

#### 3.3.2. Expected Benefits of Earnings from the Farmer’s Perspective

Let the benefit expectation of farmers when they adopt low-carbon production technologies be $U_{21}$, the benefit expectation of farmers when they do not adopt them be $U_{22}$, and the average benefit expectation be $U_{2}$, as follows.
(4)$$\begin{array}{l} U_{21}=xz\left(\theta A-\mu \eta E-C_{1}+M_{1}+r\rho \right)+x\left(1-z\right)\left(\theta A-\mu \eta E-C_{1}+M_{1}\right)+z\left(1-x\right)\left(F_{1}-C_{1}+M_{1}\right)+\left(1-x\right)\left(1-z\right)\left(F_{1}-C_{1}+M_{1}\right)\\ =xzr\rho +x\left(\theta A-\mu \eta E-C_{1}+M_{1}\right)+\left(1-x\right)\left(F_{1}-C_{1}+M_{1}\right) \end{array}$$
(5)$$\begin{array}{l} U_{22}=xz\left(M_{2}-\gamma D-C_{2}\right)+x\left(1-z\right)\left(M_{2}-\gamma D-C_{2}\right)+z\left(1-x\right)\left(M_{2}-\gamma D-C_{2}\right)+\left(1-x\right)\left(1-z\right)\left(M_{2}-\gamma D-C_{2}\right)\\ =M_{2}-\gamma D-C_{2} \end{array}$$
(6)$$\begin{array}{l} \bar{U_{2}}=U_{21}y+U_{22}\left(1-y\right)\\ =xyzr\rho +xy\left(\theta A-\mu \eta E-C_{1}+M_{1}\right)+\left(1-x\right)y\left(F_{1}-C_{1}+M_{1}\right)+\left(1-y\right)\left(M_{2}-\gamma D-C_{2}\right) \end{array}$$

#### 3.3.3. Revenue Expectations from the Consumer Perspective

Let the consumer’s benefit expectation at the time of purchase be $U_{31}$, the consumer’s benefit expectation at the time of non-purchase be $U_{32}$, and the average benefit expectation is $\bar{U_{3}}$, respectively.
(7)$$U_{31}=xy\left(M_{3}+B\varphi \right)+y\left(1-x\right)\left(M_{3}+B\varphi \right)=M_{3}+B\varphi$$
(8)$$U_{32}=xyM_{4}+x\left(1-y\right)M_{4}+y\left(1-x\right)M_{4}+\left(1-x\right)\left(1-y\right)M_{4}=M_{4}$$
(9)$$\bar{U_{3}}=U_{31}z+U_{32}\left(1-z\right)=\left(M_{3}+B\varphi \right)z+\left(1-z\right)M_{4}$$

### 3.4. Solution by Replicating the Dynamic Equation Evolutionary Stabilisation Strategy

From the above analysis, the dynamic replication equation of the government is obtained as:(10)$$F\left(x\right)=\frac{dx}{dt}=x\left(U_{11}-\bar{U_{1}}\right)=x\left(1-x\right)\left[y\left(\mu \eta E-Q-\theta A+2F_{1}\right)+\left(1-y\right)\left(\gamma D+2F_{2}-2P\beta \right)\right]$$

The dynamic replication equation for the farmers is:(11)$$F\left(y\right)=\frac{dy}{dt}=y\left(U_{21}-\bar{U_{2}}\right)=y\left(1-y\right)\left(xzr\rho +x\left(\theta A-\mu \eta E-C_{1}+M_{1}\right)+\left(1-x\right)\left(F_{1}-C_{1}+M_{1}\right)+M_{2}-\gamma D-C_{2}\right)$$

The dynamic replication equation for the consumer is:(12)$$F\left(y\right)=\frac{dz}{dt}=z\left(U_{31}-\bar{U_{3}}\right)=z\left(1-z\right)\left(M_{3}+B\varphi +M_{4}\right)$$

### 3.5. Analysis of Tripartite Evolutionary Stabilisation Strategies

From Equations (10)–(12) the system of replication dynamic equations for government, farmers and consumers is:(13)$$\left\{\begin{array}{c} F\left(x\right)=x\left(1-x\right)\left[y\left(\mu \eta E-Q-\theta A+2F_{1}\right)+\left(1-y\right)\left(\gamma D+2F_{2}-2P\beta \right)\right]\\ F\left(y\right)=y\left(1-y\right)\left(xzr\rho +x\left(\theta A-\mu \eta E-C_{1}+M_{1}\right)+\left(1-x\right)\left(F_{1}-C_{1}+M_{1}\right)+M_{2}-\gamma D-C_{2}\right)\\ F\left(z\right)=z\left(1-z\right)\left(M_{3}+B\varphi +M_{4}\right) \end{array}\right.$$

Friedman argues that the evolutionary stability strategy of a system of differential equations for a system of differential equations can be derived from the local stability analysis of the Jacobian matrix of the system, and the Jacobian matrix can be derived from (13) as follows.
(14)$$\left[\begin{array}{ccc} \left(1-2x\right)\left(y\left(\mu \eta E-Q-\theta A+2F_{1}\right)+\left(1-y\right)\left(\gamma D+2F_{2}-2P\beta \right)\right) & x\left(1-x\right)\left[\mu \eta E-Q-\theta A+2F_{1}\right] & x\left(1-x\right)\left[y\left(\mu \eta E-Q-\theta A+2F_{1}\right)+\left(1-y\right)\left(\gamma D+2F_{2}-2P\beta \right)\right]0\\ y\left(1-y\right)\left(zr\rho +\theta A-\mu \eta E-C_{1}-F_{1}+C_{1}\right) & \left(1-2y\right)\left(\begin{array}{c} xzr\rho +x\left(\theta A-\mu \eta E-C_{1}+M_{1}\right)\\ +\left(1-x\right)\left(F_{1}-C_{1}+M_{1}\right)+M_{2}-\gamma D-C_{2} \end{array}\right) & y\left(1-y\right)\left(xr\rho +x\left(\theta A-\mu \eta E-C_{1}+M_{1}\right)+\left(1-x\right)\left(F_{1}-C_{1}+M_{1}\right)+M_{2}-\gamma D-C_{2}\right)\\ z\left(1-z\right)\left(M_{3}+B\varphi +M_{4}\right) & z\left(1-z\right)\left(M_{3}+B\varphi +M_{4}\right) & \left(1-2z\right)\left(M_{3}+B\varphi +M_{4}\right) \end{array}\right]$$

In Equation (13), the eight local equilibrium points of T1(0, 0, 0), T2(0, 1, 0), T3(0, 0, 1), T4(1, 0, 0), T5(1, 1, 0), T6(1, 0, 1), T7(0, 1, 1), and T8(1, 1, 1) are obtained by putting F(x) = F(y) = F(z) = 0. The entropy values of these nodes are calculated to determine the equilibrium solutions in each nonequilibrium state. Finally, the Hamiltonian sequence corresponding to each equilibrium state and its corresponding Jacobi determinant is solved using the Lagrange multiplier method. The equilibrium point when all the eigenvalues in the Jacobi matrix are negative is the system’s evolutionary stability point (ESS).

Taking T1(0, 0, 0) as an example and substituting it into the Jacobs matrix, we can get:(15)$$\left[\begin{array}{ccc} \gamma D+2F_{2}-2P\beta & 0 & 0\\ 0 & F_{1}-C_{1}+M_{1}+M_{2}-\gamma D-C_{2} & 0\\ 0 & 0 & M_{3}+B\varphi +M_{4} \end{array}\right]$$

At this time, the eigenvalues of T1(0, 0, 0) are λ_1_ = $\gamma D+2F_{2}-2P\beta$, λ_2_ = $-C_{1}+M_{1}+M_{2}-\gamma D-C_{2}$, λ_3_ =$\;M_{3}+B\varphi +M_{4}$, Similarly, the eigenvalues of the other seven local equilibrium points can be obtained. Due to the complicated parameter settings, it is difficult to determine the evolutionary stability point by mathematical methods only, so this paper uses Matlab2021b to establish an evolutionary model and analyze the reaction strategies of the subjects of each party.

## 4. Sensitivity Analysis

From the above, it can be seen that the strategic choices of farmers, government, and consumers influence each other and vary with the benefit parameters of each party. To more intuitively represent this influence relationship, this paper combines the actual setting that the sum of the government’s penalty for cheating and the penalty for not adopting low-carbon production behavior is greater than the cost of cheating inspection, the subsidy to farmers, and the input for building carbon labeled market; the benefit of consumers not purchasing carbon labeled agricultural products is smaller than the benefit and additional benefit of consumers purchasing carbon labeled agricultural products. That is Eη+Dγ>Qμ+Aθ+Pβ; $M_{4}<M_{3}+r\varphi$, referring to the simulation parameter settings in the literature. A = 0.6, Q = 0.05, E = 1.4, D = 0.3, $C_{1}$ = 0.6, $M_{1}$ = 1, $F_{1}$= 1.5, $M_{2}$ = 0.5, $C_{2}$ = 0.6, $F_{2}$ = 0.7, P = 0.2, $M_{3}$ = 0.1, B = 0.4, r = 0.1, and $M_{4}$ = 0.04. Based on the assumptions of the above parameter variables and Equations (1)–(12), Matlab2021b was used to simulate and analyze the evolutionary stabilization strategy for the three parties of government, farmers, and consumers, and the simulation results are shown in Figure 2.

### 4.1. Effect of Government Subsidy Coefficients on the Evolution of the System

As seen in Figure 2. the greater the government subsidy, the greater the rate of evolution of the government’s strategy towards the subsidy, with no significant effect on consumers’ willingness to purchase. The growth of the government subsidy coefficient from 0 to 1 is accompanied by an increase in farmers’ willingness to adopt low-carbon production technologies. Firstly, from the perspective of government agents, the larger the government subsidy coefficient, the slower its rate of convergence to the subsidy choice, implying a slight decrease in the willingness of government subsidies, which is basically in line with reality. Secondly, from the perspective of farmers, their willingness to adopt low-carbon production technology increases as the government subsidy coefficient increases. This analysis may be because farmers will incur certain costs to learn and improve the technology by adopting low-carbon production technology. Thus, farmers will anticipate the market risk of lower yields and reduced profits for agricultural products. It is further observed that when the government subsidy coefficient is at [0, 0.75], the proportion of farmers who choose not to adopt a low-carbon production strategy is more significant than those who adopt low-carbon production. When the government subsidy coefficient is at (0.75, 1], farmers finally evolve to adopt a low-carbon production strategy. Thus, it can be seen that the government subsidy coefficient in the (0.75, 1] interval can bear the opportunity cost of farmers’ adoption of low-carbon production technology. Finally, for consumers, government subsidies for farmers’ low-carbon production practices do not differentiate between carbon-labeled agricultural products and ordinary agricultural products but merely reduce the price gap between the two, except that with government subsidies, consumers begin to pay attention to food safety source management and purchasing carbon-labeled agricultural products is the general trend. Based on the above analysis, when the government subsidy coefficient is in the (0.75, 1] range, not only do consumers quickly tend to choose to buy, but the government eventually tends to choose to subsidize. Farmers also tend to adopt low-carbon production technology, and the interests of the three parties are satisfied to different degrees.

### 4.2. Effect of Cheating Penalty Coefficient on System Evolution

As shown in Figure 3, the higher the fraud penalty coefficient, the faster the government evolves towards the subsidy strategy and the faster the farmers evolve towards the non-adoption of the low-carbon production strategy. In contrast, the purchase strategy for consumers remains almost unchanged. First, from the perspective of the government agent, the higher the penalty coefficient for cheating, the faster the government evolves toward the subsidy strategy. The reason behind analyzing this choice may be that the government’s demand for environmental management grows with the increasing economic development at the cost of ecology, and environmental regulation of ecology and agricultural carbon emissions becomes a general trend to improve the overall social welfare level. Secondly, from the farmers’ perspective, the higher the penalty coefficient for cheating, the faster the rate of evolution of farmers towards not adopting low-carbon production strategies. In order to maximize their benefits, farmers will risk violating laws and regulations to obtain government subsidies. It is further observed that when the penalty coefficient is [0.75, 1], farmers eventually choose to adopt the low-carbon production strategy. When the penalty coefficient is [0, 0.75), farmers eventually choose not to adopt the low-carbon production strategy. Thus, it can be seen that government subsidies for low-carbon production are not sufficient to compensate farmers for the cheating penalty when the cheating penalty coefficient is in the range of [0.75, 1]. Finally, for consumers, there is no significant effect on purchase intention because the government cheating penalty does not affect their utility. In summary, the coefficient of the cheating penalty is in the interval of [0.75, 1], where the three parties reach a steady state.

### 4.3. Effect of the Farmer’s Non-Production Penalty Coefficient on the Evolution of the System

As shown in Figure 4, the larger the coefficient of penalty for farmers’ non-production, the greater the rate of government evolution towards the subsidy strategy and the increase in the proportion of farmers adopting low-carbon production technologies, which still has no significant effect on consumers. First, from the perspective of the government agent, the larger the coefficient of penalty for non-production of farmers, the larger the rate of convergence to the choice of subsidies, and the penalty that farmers pay when they do not adopt low-carbon production, the government can use it for the construction of carbon-labeled agricultural markets. To force farmers to adopt low-carbon production technology production. Secondly, from the perspective of farmers, the larger the coefficient of penalty for farmers not producing, the more significant the proportion of farmers adopting low-carbon production technology increases, but in the evolutionary process, eventually choose not to produce a strategy, the possible reason is that when the coefficient of penalty for farmers not producing is small, farmers will be punished to a certain extent. However, the profits from the scale effect of traditional agricultural production can offset the cost of not low-carbon production penalty.

Conversely, if farmers stick to traditional production methods despite higher government penalties, they cannot afford the high penalty costs. Nevertheless, as time evolves, probably due to the specificity of agriculture and the low acceptance of low-carbon production technologies, farmers eventually evolve not to adopt low-carbon production methods. Finally, for consumers, there is no significant effect on their utility and no significant effect on purchase intentions due to farmers’ non-production penalties. From the above analysis, it can be seen that the change in the penalty coefficient for farmers not adopting low-carbon production has little effect on the government and consumers’ choice of subsidies and purchasing strategies, and the government can regulate farmers’ behavior by adjusting the penalty coefficient for farmers not adopting low-carbon production without violating the market law.

### 4.4. Impact of Consumer Trust Coefficient on System Evolution

As shown in Figure 5, the heterogeneity of farmers’ trust coefficients significantly affects the willingness of government subsidies and farmers’ willingness to adopt low-carbon production technologies, but not consumers. For the government, the larger the consumer trust coefficient, the slower the rate of government convergence to subsidies. The possible reason is that the active stimulus of the market mechanism is greater than the passive stimulus of government subsidies, so the more significant the consumer confidence in the market of carbon labeled agricultural products, the lower the willingness of government subsidies; for farmers, the larger the consumer trust coefficient, the slower the rate of farmer convergence to adopt low carbon production technologies, further confirming that farmers are more sensitive to For consumers, the consumer trust coefficient has no effect on strategy choice. Consumers will eventually choose to buy carbon-labeled agricultural products.

### 4.5. Effect of Additional Benefit Coefficients on the Evolution of the System

As shown in Figure 6 there is no significant difference between the government and farmers with different extra benefit coefficients for consumers, and the higher the extra benefit coefficient, the higher the rate of convergence of consumers to purchase. From the perspective of the government and farmers, the extra benefit coefficient has no effect on strategy choice, and both the government and farmers will eventually choose to subsidize low-carbon production and adopt low-carbon production technology; from the perspective of consumers, the extra benefit can promote the choice of purchasing strategy, and safer agricultural products and greener natural environment are also consumers’ irreversible consumption preferences.

### 4.6. Impact Coefficient of Willingness to Pay Premium for Carbon Labelled Agricultural Products on System Evolution

As shown in Figure 7, the greater the willingness to pay carbon labeled agricultural premium, the greater the rate of convergence of farmers to adopt low-carbon production. Conversely, there is no significant effect on the government and consumers. For the government and consumers, the willingness to pay carbon labeled agricultural products premium has no effect on the strategy choice, and both the government and consumers will eventually choose to subsidize low carbon production and purchase low carbon agricultural products; for farmers, the willingness to pay carbon labeled agricultural products premium can promote the choice of adopting low carbon production technology strategy because the willingness to pay carbon labeled agricultural products premium means that it can work directly For farmers, the willingness to pay for carbon labeled agricultural products can promote the adoption of low-carbon production technology strategies because the willingness to pay for carbon labeled agricultural products means that consumer preferences for carbon labeled agricultural products can be directly realized. Farmers can be directly stimulated to adopt low-carbon production on the market side.

## 5. Discussion

The academic contribution of this paper is mainly reflected in the following three aspects: first, compared with the previous studies on food safety supervision mechanism, most of them focused on the production and operation entities [47] or government supervision departments [48], this paper has more innovative research on the impact of the game between farmers, the government and consumers [49], which is an important source of food safety, on the public environment [50], which helps to make up for the lack of relevant literature; Second, this study reveals the possible mechanism by which the government and consumers influence the low-carbon production behavior of farmers, which helps to understand the complex relationship between whether farmers implement low-carbon production behavior and policy regulation and market stimulation; Third, this study is also instructive to explore how policy regulation [51] and market stimulation can improve the situation of public goods when public goods are damaged and need to be protected in public economics [52].

The findings of this paper have important policy implications for managing the low-carbon production behavior of farmers at the source of food safety: the institutional construction of carbon-labeled agricultural markets needs to be further improved. The market incentives of carbon labels for low-carbon production behavior of farmers need to be enhanced, and the leading role of market mechanisms in managing food safety at the source needs to be gradually established. Although both policy regulation and market stimulation can achieve the goal of cleaning up the agricultural production ecosystem by increasing farmers’ expected returns, the significant advantage of carbon labeling agricultural markets over policy-regulated low-carbon production tools is that they can reduce the cost of the agricultural production ecosystem management through market trading mechanisms, thereby minimizing the cost of management under the established food safety source management objective. The study concludes that although carbon-labeled agricultural markets have significant pro-environmental behavioral effects on farmers’ low-carbon production behavior, carbon labeling plays a limited role. Although the government can also guide farmers to adopt low-carbon production through policy regulation, the lack of a functional market mechanism will make the carbon label agricultural market vain, and it will be challenging to reduce the ecological and environmental management costs of agricultural production through the market mechanism. Therefore, it is necessary to derive the types of products in the carbon market, strengthen the construction of the carbon label agricultural market system, improve the statistics of carbon emission data of agricultural products, verification and public announcement of low-carbon production of agricultural products.

Of course, there are certain shortcomings in this paper, mainly in how to maximize the encouragement of farmers to adopt low-carbon production and invest in the camp of building carbon labeled agricultural products market to govern the source of food safety, which still needs the guarantee of the market mechanism as well as government regulation mechanism, and how to design a long-term and stable mechanism to realize the improvement of the source of food safety governance is the focus of the subsequent research.

This section may be divided by subheadings. It should provide a concise and precise description of the experimental results, their interpretation, as well as the experimental conclusions that can be drawn.

## 6. Conclusions

This paper takes the government, farmers, and consumers as the main subjects of research in food safety governance at source and discusses the impact of their strategic choices under different policy scenarios in terms of content and intensity.

The conclusions of this study are: (i) the government subsidy coefficient, farmer non-production penalty coefficient, consumer trust coefficient, different benefit coefficient, and willingness to pay carbon labeled agricultural products premium coefficient are positively correlated with farmers’ adoption of low carbon production behavior, the stronger the incentive, the greater the probability of farmers’ adoption of low carbon production behavior; the fraudulent subsidy penalty coefficient is negatively correlated with farmers’ adoption of low carbon production behavior, the smaller the value, the greater the probability of farmers’ adoption of low carbon production behavior. The smaller the value, the greater the probability that farmers will adopt low-carbon production behavior; (ii) The sensitivity of farmers to government regulation policies is: fraud penalty coefficient > government subsidy coefficient > farmers do not produce penalty coefficient, so special attention needs to be paid to the supervision of fraudulent farmers, farmers’ sensitivity to direct market stimulation is: consumer trust coefficient > carbon labeling agricultural premium willingness to pay influence coefficient, the additional benefit coefficient has no significant impact on farmers’ decisions; and (iii) China’s food safety source management mainly relies on the government’s policy intervention in the early stage and the establishment of a market stimulation mechanism for carbon labeled agricultural products in the later stage in order to achieve long-term stable and effective food safety source management.

## Figures and Tables

**Figure 1 ijerph-19-12211-f001:**
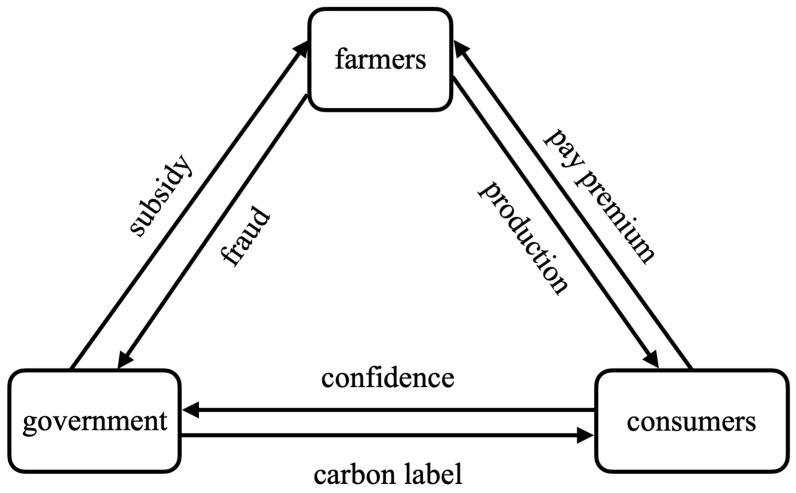
Map of incentives and integration mechanisms for food safety source governance.

**Figure 2 ijerph-19-12211-f002:**
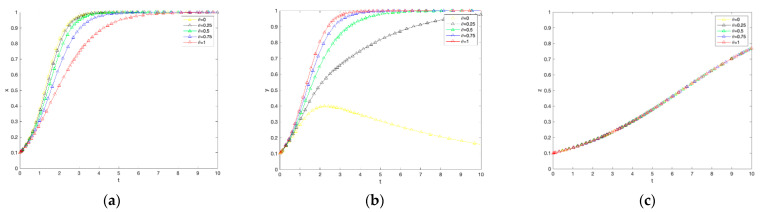
The impact of changes in government subsidy coefficients on government subsidies, farmer adoption, and consumer purchases. (**a**) Government response. (**b**) Farmers’ response. (**c**) Consumer reaction.

**Figure 3 ijerph-19-12211-f003:**
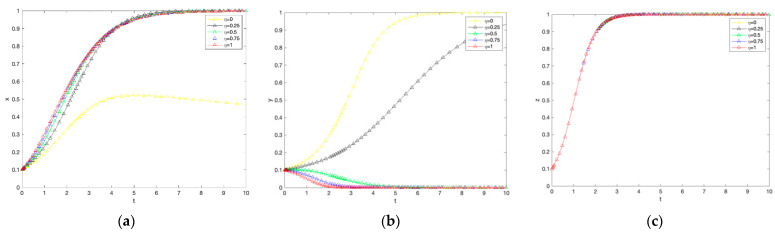
The impact of changes in the penalty coefficient on government subsidies, farmer adoption, and consumer purchases. (**a**) Government response. (**b**) Farmers’ response. (**c**) Consumer reaction.

**Figure 4 ijerph-19-12211-f004:**
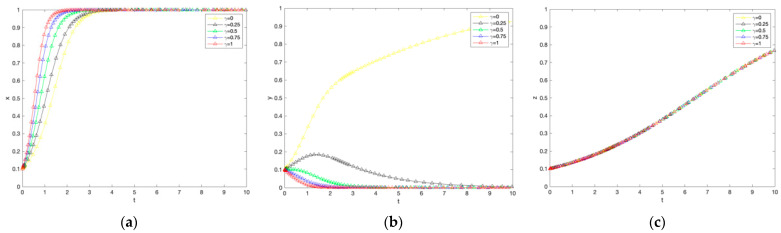
The impact of changes in the penalty coefficient for farmers not producing on government subsidies, farmer adoption, and consumer purchases. (**a**) Government response. (**b**) Farmers’ response. (**c**) Consumer reaction.

**Figure 5 ijerph-19-12211-f005:**
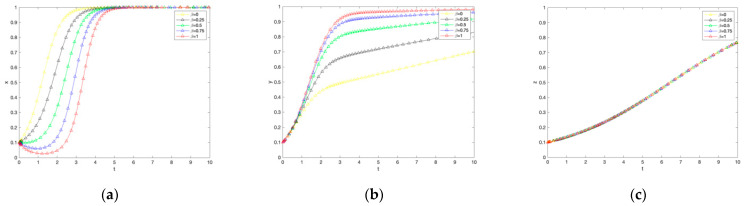
The impact of changes in consumer trust coefficients on government subsidies, farmer adoption, and consumer purchases. (**a**) Government response. (**b**) Farmers’ response. (**c**) Consumer reaction.

**Figure 6 ijerph-19-12211-f006:**
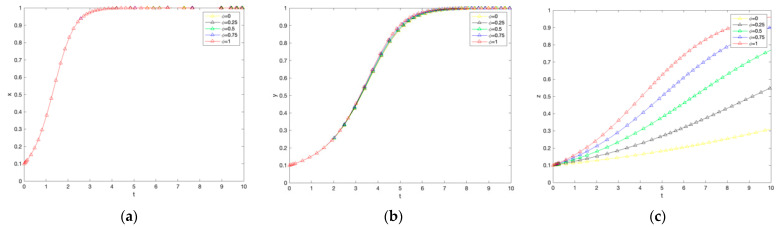
The impact of changes in the additional income coefficient on government subsidies, farmer adoption, and consumer purchases. (**a**) Government response. (**b**) Farmers’ response. (**c**) Consumer reaction.

**Figure 7 ijerph-19-12211-f007:**
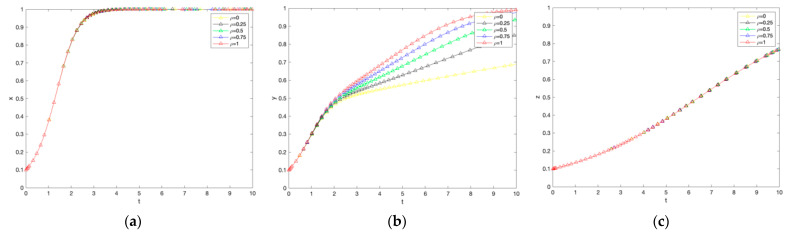
The impact of changes in the influence coefficient of the willingness to pay premiums for carbon-labelled agricultural products on government subsidies, farmer adoption, and consumer purchases. (**a**) Government response. (**b**) Farmers’ response. (**c**) Consumer reaction.

**Table 1 ijerph-19-12211-t001:** Parameter descriptions.

Parameters	Descriptions
A	Government’s low-carbon subsidies to farmers
θ	subsidy coefficient
Q	Cost of government inspection of whether farmers have fraud
μ	Probability of farmers cheating
E	Government penalties for farmers
η	Fraud penalty factor
D	Government Punishment on Farmers’ Non-Low Carbon Production
γ	penalty coefficient
$C_{1}$	Low-carbon production cost of farmers
$M_{1}$	Sales income of carbon label agricultural products
$F_{1}$	Low carbon social benefits
$C_{2}$	Loss of carbon label safety of agricultural products market benefits
$M_{2}$	General sales income
$F_{2}$	General social benefits
P	Construction of Carbon Label Agricultural Products Market Investment
β	Consumer trust coefficient
$M_{3}$	Consumers’ carbon label agricultural products income
B	Consumers’ extra environmental benefits
φ	Extra income coefficient
r	Additional carbon label agricultural income
ρ	Impact coefficient of willingness to pay for carbon label agricultural products premium on income
$M_{4}$	Consumer ordinary income

**Table 2 ijerph-19-12211-t002:** The benefit matrix of the farmer-consumer evolutionary game under the background of government subsidies.

		Consumers	
		Purchase (z)	No Purchase (1 − z)
**Farmers**	**Adopted** **(y)**	$\mu \eta E-Q-\theta A+F_{1}$	$\mu \eta E-Q-\theta A+F_{1}$
		$\theta A-\mu \eta E-C_{1}+M_{1}+r\rho$	$\theta A-\mu \eta E-C_{1}+M_{1}$
		$M_{3}+B\varphi$	$M_{4}$
	**Do not use** **(1 − y)**	$\gamma D+F_{2}-P\beta$	$\gamma D+F_{2}-P\beta$
		$M_{2}-\gamma D-C_{2}$	$M_{2}-\gamma D-C_{2}$
		0	$M_{4}$

**Table 3 ijerph-19-12211-t003:** The income matrix of the farmer-consumer evolutionary game in the context of government non-subsidy.

		Consumers	
		Purchase (z)	No Purchase (1 − z)
**Farmers**	**Adopted** **(y)**	$F_{1}$	$F_{1}$
		$F_{1}-C_{1}+M_{1}$	$F_{1}-C_{1}+M_{1}$
		$M_{3}+B\varphi$	$M_{4}$
	**Do not use** **(1 − y)**	$F_{2}-P\beta$	$F_{2}-P\beta$
		$M_{2}-\gamma D-C_{2}$	$M_{2}-\gamma D-C_{2}$
		0	$M_{4}$

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
