# Peer review of "The Evolutionary Game Analysis of Low Carbon Production Behaviour of Farmers, Government and Consumers in Food Safety Source Governance"

_ijerph, 2022, doi:10.3390/ijerph191912211_

Round 1
Reviewer 1 Report
The article reveals that government subsidy coefficient, farmers' penalty coefficient for not producing, consumer trust coefficient and willingness to pay carbon labelled agricultural products premium are positively related to farmers' adoption of low-carbon production behaviour, and fraud penalty coefficient and farmers' cost of adopting low-carbon production technology are negatively related; Food safety source management mainly relies on government policy intervention in the early stage, and the establishment of a market incentive mechanism for carbon labelled agricultural products in the later stage can achieve long-term stable and effective food safety source. The analysis process is comprehensive, good organized, large amount of information and so on. Minor revision can be published in International Journal of Environmental Research and Public Health. However, there are some major issues need to be improved:
1. Abstract: The readability needs to be further improved; For example, the last two repetitive words can be merged.
2. Introduction: Some contents are not marked with references, such as the first paragraph;
3. Theoretical analytical framework: Whether this part can be put in the preface?
4. Evolutionary Game Model: What is the difference between this part and the Materials and Methods? Can the latter be replaced?
5. Sensitivity Analysis: Is this partly similar to the results and analysis?
6. Discussion: The research results of this paper and the same kind at home and abroad clarify the innovation point of the paper
7.
8. Conclusions:The content is too little, so we need to be added;
9. References: Authors should revise the format of reference according to IJERPH Journal..
Author Response
Dear Reviewers:
I’m writing to thank you very much for returning to us the reviews of our manuscript, ijerph-1861750 This paper was submitted to your journal on the 27 July 2022 At this stage, we would like to thank both you for spending time to work on our paper. We very much appreciate all of the comments received in review.
We have studied comments carefully and have made correction which we hope meet with approval. Revised portion are marked in red in the revised manuscript.A point-by-point response to the reviewers' comments is attached below for your review.
Thank you for your time with our work. We look forward to hearing from you.
We will look forward to hearing from you.
Yours Sincerely,
[Yayan Xie]
[College of Economics and Trade Xinjiang Agricultural University]
[320213469@xjau.edu.cn]

Reviewer 2 Report
This manuscript deals with a relevant and contemporary issue: the game among government, farmers and consumers in face of the Low Carbon Production issue.
The authors explore the evolutionary game theory to modelize and simulate the interactions and behavior of government, farmers and consumers and analyze the dynamic evolutionary process and evolutionary stable strategies of these three stakeholders.
The research issue and objectives are clear and coherently stated.
The analytical framework and the method are clear, coherent and soundly stated.
The results are clear and soundly presented and discussed. The reserach results are provocative.
The conclusions are pertinent and supported by the research results. The public policies and managerial implications are pertinet and provocative.
Nevertheless, some points seems worth revisiting:
1 - In Figure 1 the arrow from consumers to farmers couldn' t be "pay premium" instead of "buy"?
2 - In Figure 1 the arrow from farmers to goverment couldn' t be "fraud" instead of "punishment"?
3 - In Section 2, from line 178 to line 189, it is presented Hypotheses H1 and H1. Nevertheless, in Section 3, from line 192 to li 224, it is presented a discussion about H1, H2, H3 and H4. Are there H3 and H4? If OK, they shoud be presented in Section 2.
4 - In Section 5 - Discussion - it seems pertinent to discuss about the Hypotheses "confirmation" and/or "rejection".
5 - The manuscript title could be added "Goverment" and "Consumers". Suggestion : "The Evolutionary Game Analysis of Low Carbon Production Behaviour of Farmers, Government and Consumers in Food Safety Source Governance"
Author Response
Dear Reviewer:
I’m writing to thank you very much for returning to us the reviews of our manuscript, ijerph-1861750 This paper was submitted to your journal on the 27 July 2022 At this stage, we would like to thank both you for spending time to work on our paper. We very much appreciate all of the comments received in review.
We have studied comments carefully and have made correction which we hope meet with approval. Revised portion are marked in red in the revised manuscript.A point-by-point response to the reviewers' comments is attached below for your review.
Thank you for your time with our work. We look forward to hearing from you.
We will look forward to hearing from you.
Yours Sincerely,
[Yayan Xie]
[College of Economics and Trade Xinjiang Agricultural University]
[320213469@xjau.edu.cn]

Reviewer 3 Report
This manuscript contains information related the low carbon production behavior of farmers and have importance for environment and food safety. The authors noted that the institutions should be more involved to promote carbon-labeled agricultural markets.
There are some shortcomings:
- In introduction section author should describe what are the means to apply low-carbon production technology
- Authors should more emphasize the relation of paper with public health
- In Discussion authors should compare/analyses the results of paper with other studies from international literature.
Author Response

(The authors gave the same response as above.)

Round 2
Reviewer 3 Report
The authors of the paper responded to all comments/suggestions. The quality of the manuscript has been improved. Manuscript shows practical and scientific importance. The work may be accepted for publication.